# Easy Intra-Operative Localization of Pulmonary Nodules during Uniportal Video-Assisted Thoracoscopy: Experience with Hydrogel Plugs at Our Institution

**DOI:** 10.3390/medsci10040054

**Published:** 2022-09-23

**Authors:** Filippo Longo, Rosario Francesco Grasso, Giovanni Tacchi, Luca Frasca, Eliodoro Faiella, Pierfilippo Crucitti

**Affiliations:** 1Unit of Thoracic Surgery, Department of Medicine and Surgery, Campus Bio-Medico di Roma University, Via Alvaro del Portillo, 00128 Rome, Italy; 2Department of Medicine and Surgery, Section of Interventional Radiology-Imaging Center, Campus Bio-Medico di Roma University, Via Alvaro del Portillo, 00128 Rome, Italy; 3PhD Course in Microbiology, Immunology, Infectious Diseases and Transplants (MIMIT), University of Rome, Tor Vergata, 00133 Rome, Italy; 4Department of Radiology, Sant’Anna Hospital via Ravona, San Fermo della Battaglia, 22042 Como, Italy

**Keywords:** lung cancer, ground glass opacity, hydrogel plug, uniportal VATS

## Abstract

Background: The diffusion of lung cancer screening programs has increased the detection of both solid and ground-glass opacity (GGO) sub-centimetric lesions, leading to the necessity for histological diagnoses. A percutaneous CT-guided biopsy may be challenging, thus making surgical excision a valid diagnostic alternative. CT-guided hydrogel plug deployment (BioSentry^®^) was recently proposed to simplify intraoperative nodule localization. Here, we report our initial experience. Methods: We evaluated 62 patients with single, small, peripheral, non-subpleural pulmonary GGO that was suspicious for cancer. All lesions were preoperatively marked, using CT-guidance, with a hydrogel plug (BioSentry^®^). Then, a uniportal video-assisted thoracoscopy (uniVATS) wedge resection was performed. If cancer was confirmed at the frozen section, a major lung resection was then performed. The study’s end points were the rates of intraoperative localization and of successful resection. Results: The hydrogel plug was correctly placed in 54 of the 62 cases, leading to an effective resection of the target lesion. In the remaining eight cases, the plug was displaced, and so the identification of pleural erosions due to the previous percutaneous procedure guided the resection. The uniVATS resection success rate was 98.3%. Conclusions: CT-guided hydrogel plug placement allowed for the successful detection of lung GGOs and resection with the uniVATS approach. This device allowed us to obtain lung cancer diagnoses and successfully treat 85.4% of cases.

## 1. Introduction

To reduce lung cancer-related mortality, specific screening programs have developed in recent decades [1,2,3,4,5,6]. As a consequence, there has been an increased detection of indeterminate millimetric solid nodules and ground-glass opacity (GGO), leading to the development of new protocols for the radiologic follow-up of such pulmonary findings [7,8]. Variations in a nodule’s dimensions or morphology usually requires a histological analysis in order to potentially plan treatment. In this setting, even though a CT-guided fine needle core-biopsy (FNCB) may represent an ideal approach, it may be challenging due to the nature and small diameters of the target lesions. Minimally invasive surgical biopsies (VATS) have proven to be a valid alternative to percutaneous approaches by allowing a wedge resection; nevertheless, intraoperative nodule identification may be difficult due to the impossibility of employing finger palpation or indirect palpation with instruments during VATS [9]. 

In recent years, several methods have been proposed to mark peripheral, small, solid nodules and GGO before surgical excision. Overall, hook-wire, microcoils, and lipiodol have been tested and reported on by different authors [10,11,12,13]. These marking techniques are burdened by minor and major complications, including: pneumothorax and parenchymal bleeding, device dislodgement (especially for hook-wire), changings in histology (a microcoil can alter a nodule’s structure), and high costs, such as the requirement for fluoroscopic assistance (lipiodol) [14,15,16,17,18]. Recently, a CT-guided hydrogel plug deployment was proposed for peripheral lung nodule and GGO preoperative localization. Initially used to reduce pneumothorax incidence after lung FNCB [19], some authors have proposed using this device to mark lung lesions before performing surgery [20]. 

Here, we report on our initial experience with hydrogel plug (BioSentry^®^, AngioDynamics, Queensbury, NY, USA) application in patients undergoing uniportal VATS resections for GGOs, which are the most challenging lung lesions to deal with.

## 2. Materials and Methods

### 2.1. Patients’ Characteristics

Between February 2019 and August 2021, we retrospectively evaluated data from 62 patients (33 females and 29 males; mean age of 65.2 years, 7.4 SD) affected by single, small, peripheral, non-subpleural pulmonary GGOs that were managed by surgical excision at our institution (the patients’ lesions’ characteristics are listed in Table 1). All patients had peripheral lesions characterized by dimensional growth or suspicious morphology. The inclusion criteria were: lesion unfit for CT-guided FNCB for small dimensions [21,22] and lesion with a previous non-diagnostic biopsy. 

### 2.2. Plug Deployment Procedure

All lesions were preoperatively marked by the deployment of a CT-guided hydrogel plug (BioSentry^®^, AngioDynamics, Queensbury, NY, USA). This device is made of absorbable polyethylene glycol (PEG) hydrogel, which has a secure profile for what is considered toxicity and mutagenicity. All the procedures were performed with a CT navigation system that had been validated in previous studies (SIRIO, Masmec, Bari, Italy) [23]. All the patients signed a written informed consent before undergoing the plug deployment procedure. Ethics committee approval was not requested for several reasons: the device is routinely used by radiologists to control haemostasis and aerostasis in patients undergoing a lung biopsy, its effectiveness has been already demonstrated [24,25,26], and, according to the Food and Drug Administration, this device is indicated to mark lung nodules for visualization during surgical resection [27].

A 64-MDCT scanner (Somatom Sensation, Siemens, Forchheim, Germany) was used in all cases, applying a low-dose radiation protocol. The procedures were performed under local anesthesia (10–20 mL mepivacaine hydrochloride 2% on the parietal surface of the pleura). Mild sedation was obtained by the administration of 1–2 mg of midazolam. 

Once the low-dose CT examination was carried out, the images were sent to the navigation system in DICOM format. At this point, using the guidance of the navigation system in spontaneous breathing, since the navigator is equipped with a correction of the respiratory movements, a 19 G needle was introduced at the level of the known target. The Biosentry system was then released according to the manufacturer’s instructions (Figure 1). A control CT scan was performed at the end of the procedure to detect any complications (e.g., pneumothorax, pleural effusion, or bleeding). Though the plug is not radiopaque, it can sometimes be seen on the control CT scan (Figure 2).

### 2.3. Surgical Approach

After the radiological procedure, a patient was returned ito the ward and surgery was scheduled for the same day of plug deployment, though at least 3 h later (the time required for the plug’s expansion). We performed a uniVATS procedure under general anesthesia according to Gonzales Rivas’ approach [28]. After the lung collapse, visceral pleura were carefully inspected until direct visualization of the hydrogel plug was confirmed. Once the marked lesion was identified, a large-ring forceps was used to remove the lesion with the surrounding lung parenchyma with a margin distance of 2 cm and a wedge resection was performed (Figure 3). All lung specimens underwent frozen section examination in order to assess the lesions’ excision and its nature (benignant vs malignant). If a malignant cancer was confirmed, we proceeded to a major lung resection, and the type of resection (segmentectomy or lobectomy) depended mainly on the site of the lesion, as well as the percentage of solid component of the lesion (segmentectomy if it was less than 50% of the GGO). The study’s end points were the rate of intraoperative localization of marked lesions and the rate of successful uniVATS resections.

## 3. Results

The hydrogel plug marking was successfully performed for all patients. The mean time to perform the procedure was 18.9 min (SD 5.6). No intraparenchymal bleeding was registered. One patient developed a thoracic wall hematoma which did not delay the surgical schedule. One patient developed a pneumothorax that did not require a chest tube placement. One patient had surgical resection delayed because of evidence of a new lung consolidation suspicious for COVID-19 pneumonia during the plug deployment procedure. This patient was clinically asymptomatic but underwent two diagnostic swabs for a COVID-19 polymerase chain reaction, both of were negative. After one month, we performed the uniVATS resection and the plug was still correctly located in the lung parenchyma (Figure 3).

The marking procedure allowed the surgeons to intra-operatively correctly identify the nodule in 54 cases (87.1%), and this resulted in effective resections of the target lesions. In seven cases (11.2%), the plug was displaced in the nearby pleural space. For these patients, the visualization of the parietal/visceral pleural puncture site allowed us to localize the target lesion (Figure 4). One case (1.7%) of a right upper lobe GGO resulted in a non-diagnostic lung excision.

Among all the resections, 53 specimens (85.4%) were positive for lung cancer at the frozen section. In these patients, 20 segmentectomies and 33 lobectomies were carried out using the uniVATS approach. The remaining nine specimens (14.6%) were benign lesions at the frozen section. One case resulted in a non-diagnostic lung excision (the patient was sent for a radiological follow-up). The successful uniVATS resection rate was 98.3%. The final histology confirmed all the frozen section reports. The histological details are reported in Table 2.

## 4. Discussion

In recent decades, lung cancer screening programs have increased the detection rate of asymptomatic lung nodules, both solid and subsolid. When meeting certain characteristics, these lesions may require histopathological definition that may be challenging, in particular, for GGOs.

According to recent guidelines, pure GGOs growing in dimensions, those developing a solid component or persistent partially solid nodules with solid portion ≥6 mm, or part-solid GGOs of >15 mm at first finding require an histological examination [8,29] because of the high probability of an invasive component, which is typically an adenocarcinoma [30]. Because of their structure and the (usually) small size of the internal cancerized solid component, percutaneous biopsies may be challenging, with high rates of false negative histological results. In a patient such as this, a surgical biopsy represents a valid alternative to a percutaneous biopsy. 

GGOs are considered the most challenging lesions to treat surgically, both with a minimally invasive procedure and with an open technique, mainly because of the difficulty in finding them intraoperatively due to their unpalpable consistencies. In this scenario, several approaches have been reported for making their intra-operative localizations and resections easier, but none of them have resulted in a gold standard for this purpose. 

A recent meta-analysis identified several complication rates related to the main lung marking procedures currently performed worldwide, consisting of hook-wire, microcoils, and lipiodol placement. In this analysis, pneumothorax was observed in 35, 16% and 27% of the marking procedures, respectively, and pulmonary bleeding rates were 16%, 6%, and 10% respectively. Regarding subsequent VATS identification and successful resections, rates of 96% for the hook-wire, 97% for microcoils, and 99% for lipiodol placements were reported. Compared with these most-common marking approaches, we observed a similar efficacy in terms of successful resection rates. Conversely, our data are not charged by a similar complication’s burden. On the other hand, the hydrogel plug dislodgement’s rate in our study is slightly higher than that reported for the CT-guided hook-wire and microcoil techniques (11.2% vs. 2–9%) [17]. This is likely due to the learning curve of our experience with this device, as well as our small sample size.

The BioSentry^®^ hydrogel plug placement has previously been investigated for preventing pneumothorax after FNCB procedures, and it has been recently proposed for marking lung lesions. In particular, Imperatori et al. reported on a series of 27 patients who underwent this marking procedure [31]. They performed three-port VATS procedures with a flexible surgical schedule, and the radiologic procedure and the resection were not always performed on the same day. They reported an 11% plug dislodgement rate, though 100% of their resections were successful. They had only one patient with symptomatic pneumothorax who required a chest tube drainage. Our experience with this incoming device (i.e., our successful VATS resection rate of 98.3%) is consistent with the effectiveness, dislodgment, and complication rates reported. Distinctly, our series is only characterized by GGOs, the most challenging lung lesions to deal with. A lung cancer diagnosis was demonstrated in 85.4% of all the GGOs analyzed in this study, and this is a significant result in this setting because we reduced the exposure to follow-up CT radiation, the need for further biopsies, and the associated psychological stress for patients. Moreover, in today’s cost-conscious health care environment, we carried out both plug placement procedures and surgical excisions in the same day, during the same hospitalization period, proving the feasibility of both procedures and thus reducing overall costs. Our success is also related to the multidisciplinary approach of our institution, which allows us to share our different areas of expertise in order to deal with challenging cases [32].

## 5. Conclusions

To the best of our knowledge, this is the first report on lung cancer resections using the uniVATS approach for pulmonary GGOs previously marked with a hydrogel plug (BioSentry^®^).

Despite that this is an initial experience with a small sample of patients, our results confirm the feasibility of both mentioned procedures, proving safety for patients. Finally, we confirmed the possibility of managing GGOs, the most challenging of lung lesions, with this incoming device, thus changing the clinical history of patients and achieving a lung cancer diagnosis and treatment in 85.4% of cases, saving the effort of prolonged follow-ups.

This paper recognizes that it has limitations. First, it is a retrospective, not-randomized study. Second, we analyzed a narrow sample size and we did not have a control group. 

## Figures and Tables

**Figure 1 medsci-10-00054-f001:**
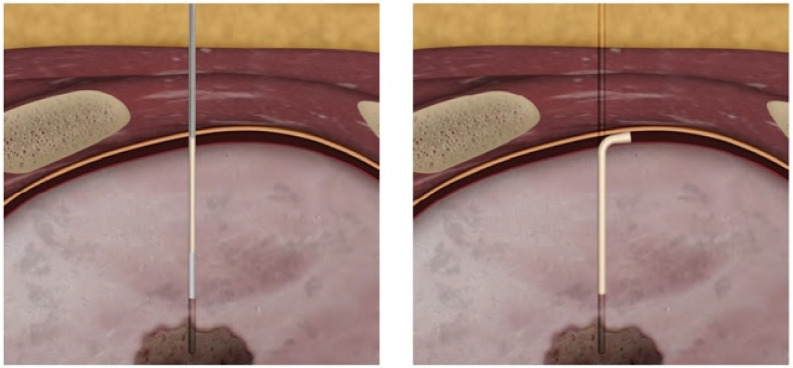
A cartoon slide showing the plug deployment.

**Figure 2 medsci-10-00054-f002:**
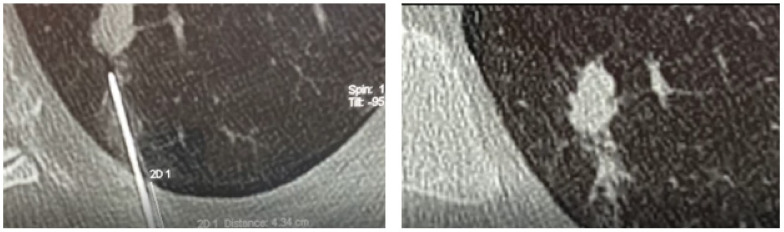
A CT scan showing the plug deployment.

**Figure 3 medsci-10-00054-f003:**
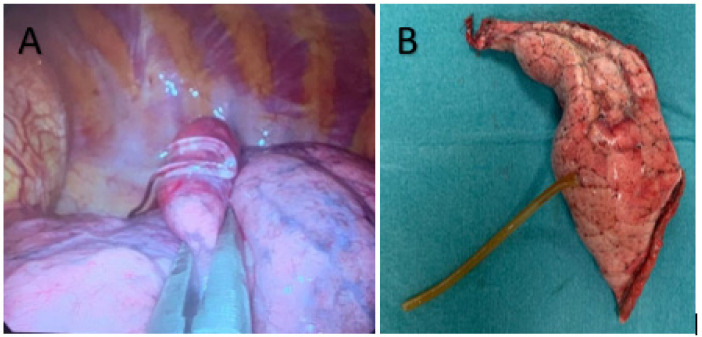
(**A**) Hydrogel plug sealed into lung parenchyma (uniVATS vision). (**B**) Lung specimen with hydrogel plug previously deployed (1 month before surgery).

**Figure 4 medsci-10-00054-f004:**
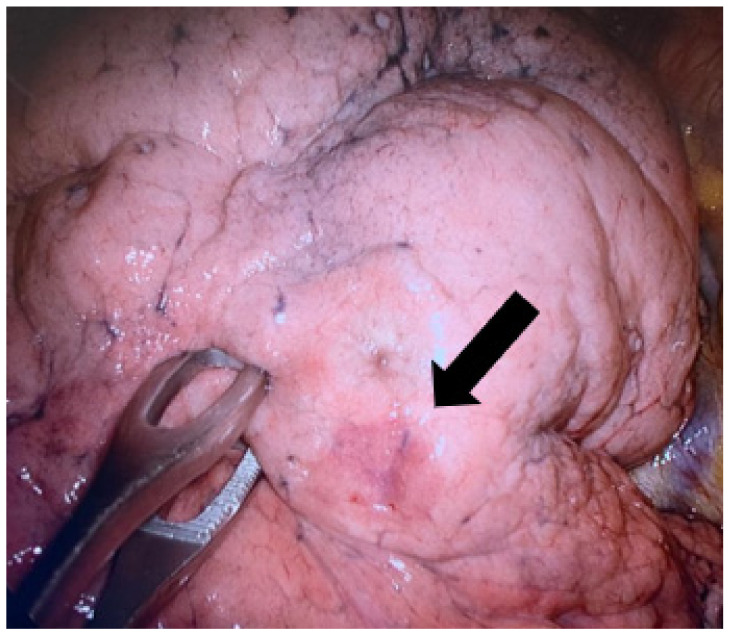
A case of hydrogel plug dislodgment. The arrow indicates the subpleural suffusion nearby the percutaneous lung puncture. Arrow: outlines the subpleural suffusion.

**Table 1 medsci-10-00054-t001:** Lesions’ Characteristics.

	Value
Radiological dimension	12.9 ± 4.8 mm
Localization	RUL, *n* = 21 (33.8%) RLL, *n* = 16 (25.8%) ML, *n* = 2 (3.2%) LLL, *n* = 13 (21%) LUL, *n* = 10 (16.2%)
Indication for marking	Suspicious morphology/growth, *n* = 48 (77.4%) Previous not-diagnostic FNCB, *n* = 14 (22.6%)
Surgical procedure	Lobectomy, *n* = 33 (53.2%) Segmentectomy, *n* = 20 (32.2%) Wedge resection, *n* = 9 (14.6%)

**Table 2 medsci-10-00054-t002:** Overall histology.

	Staging
Malignant, *n* = 53 (85.4%) Adenocarcinoma, *n* = 51 (82.2%) Large cell carcinoma, *n* = 1 (1.6%) Typical carcinoid, *n* = 1 (1.6%) Benign, *n* = 9 (14.6%) Chronic flogosis	Stage IA 1, *n* = 33 (62.3%) Stage IA 2, *n* = 15 (28.4%) Stage IA 3, *n* = 3 (5.6%) Stage II B (N1 positive), *n* = 2 (3.7%)

## Data Availability

The data presented in this study are available on request from the corresponding author. The data are not publicly available due to privacy restrictions.

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
