# Peer review of "Easy Intra-Operative Localization of Pulmonary Nodules during Uniportal Video-Assisted Thoracoscopy: Experience with Hydrogel Plugs at Our Institution"

_medsci, 2022, doi:10.3390/medsci10040054_

Round 1

Reviewer 1 Report

Dear Authors,

Thank you for giving the opportunity to review the article titled "Easy intra-operative localization of pulmonary nodules during Uniportal Video Assisted Thoracoscopy: experience with Hydrogel Plug at our Institution".

I read the article with interest and attention. I would say it is a well written article with its design, methodology, findings and discussion.

Best Regards

Author Response

Dear Colleague,

thank you for your positive revision.

I hope we will have new chance for cooperation

Best regards.

Dr Giovanni Tacchi

Reviewer 2 Report

Comment

1. CT images after placing the Plug is helpful to understand the procedure for readers. Please add the CT images, if available.

Reviewer 3 Report

The present article entitled, “Easy intra-operative localization of pulmonary nodules during Uniportal Video Assisted Thoracoscopy: experience with Hydrogel Plug at our Institution.” submitted by Crucitti et.al., reports uniVATS approach in the lung cancer resections aimed at pulmonary GGOs heretofore marked by hydrogel plug (Bio- Sentry) claims to save the exertion of a lengthy follow-ups. Although this study has relevance in cancer studies, The sample size could be increased along with valid controls.

Author Response

Dear colleague,

our paper reports preliminary results with this device. We will enlarge the sample size within the next years providing a control group in order to verify our current statement.

Thank you for your cooperation and advices. Best regards.

Dr Tacchi Giovanni